# Critical Appraisal of Advanced Glycation End Products (AGEs) and Circulating Soluble Receptors for Advanced Glycation End Products (sRAGE) as a Predictive Biomarkers for Cardiovascular Disease in Hemodialysis Patients

**DOI:** 10.3390/medsci6020038

**Published:** 2018-05-22

**Authors:** Adel M. A. Assiri, Hala F. M. Kamel, Abeer A. ALrefai

**Affiliations:** 1Biochemistry Department, Faculty of Medicine, Umm Al-Qura University, Makah 21955, Saudi Arabia; aassiri64@hotmail.com (A.M.A.A.); kamelhala@msn.com (H.F.M.K.); 2Medical Biochemistry Department, Faculty of Medicine, Ain Shams University, Cairo 11566, Egypt; 3Medical Biochemistry Department, Faculty of Medicine, Menoufia University, Shebin Al-Kom 32511, Egypt

**Keywords:** advanced glycation end products, soluble receptor advanced glycation end products, chronic kidney diseases, cardiovascular diseases, diabetes mellitus

## Abstract

The interaction of advanced glycation end products (AGE) and their receptors promote vascular complications of diabetes in hemodialysis (HD) patients. The soluble form of the receptor for the advanced glycation end-products (sRAGE) has been studied as a vascular biomarker in various diseases with controversial results. Our aim was to evaluate the association of the serum levels of the AGEs and their receptor sRAGE with cardiovascular disease (CVD) and the cardiovascular risk factors among HD patients. There were 130 HD patients and 80 age and gender matched control subjects were involved; 31.5% of the HD group were diabetic, which was an underlying cause of renal impairment; 36.1% had CVD, which was comprising 44.7% of diabetics and 55.3% of non-diabetic patients. The AGEs and sRAGE were assessed by enzyme linked immunosorbent assay (ELISA). In addition, the lipid profile, glycemic indices, pre-dialysis renal function tests, and hemoglobin % (Hb) were evaluated. The results show that the circulating AGEs and sRAGE levels were significantly higher in the HD patients. Those with underlying diabetes displayed higher sRAGE levels, which were positively correlated with hyperglycemia, HbA1C, and total cholesterol (TC). The HD patients with an increased serum sRAGE exhibited more cardiovascular risk factors (hypercholesterolemia and anemia) with a high prevalence of CVD. Using a linear regression analysis, we found a significant association of sRAGE with CVD and TC among HD patients, regardless of whether associating diabetes was an underlying cause of renal impairment. Overall, the HD patients displayed significantly higher serum AGEs with a concomitant increase in the circulating sRAGE levels, mainly in the diabetic HD, which were significantly associated with the CVD (independent predictors) and CV risk factors (hypercholesterolemia), mainly sRAGEs, regardless of the underlying diabetes mellitus. This highlights the prognostic role of AGEs and sRAGE in HD patients regardless of underlying cause in order to predict the risk for CVD.

## 1. Introduction

Chronic kidney disease (CKD) is a worldwide health problem of increasing incidence and prevalence, which is associated with the high cost of therapy and poor prognosis [1]. Patients with CKD, in particular those on dialysis, experience an extremely high cardiovascular rate of morbidity and mortality, which cannot be explained by traditional risk factors alone [2]. In uremia, the increased inflammation and increased oxidative stress may contribute to atherosclerosis [3]. In addition to the classical mechanisms, advanced glycation end products (AGEs), a large heterogeneous group that is formed by the irreversible non-enzymatic glycation and oxidative reactions, may contribute to vascular damage [4]. Aging, diabetes, and chronic renal failure has induced an abundant accumulation of AGEs. In chronic renal failure, the decreased excretion mechanism is induced by the AGEs accumulation in the tissue, where they cross-link with collagen thereby increasing vascular stiffness [5]. Recently, Koska and Associates indicated that higher levels of selected AGE were associated with increased incidences of cardiovascular disease (CVD) [6]. In addition, Cassese et al. indicated that the AGEs’ impairment of insulin action in the muscle might have been mediated by the formation of a multimolecular complex, including RAGE (receptors for the advanced glycation end products)/IRS-1 (Insulin Receptor Substrate-1)/Src and PKC (Protein Kinase C) [7]. AGEs interact with their multi-ligand member of the immunoglobulin superfamily receptor. The receptors for the advanced glycation end products (RAGE), on endothelial cells, further inhibit the endothelial nitric oxide synthase, thereby contributing to endothelial dysfunction [8], atherosclerotic processes, and a variety of microvascular and macrovascular complications [9]. AGEs contribute to endothelial dysfunction by increasing the vascular deposition of the oxidized low-density lipoproteins (LDL), reducing nitric oxide concentration, enhancing oxidative stress, and causing abundant macrophage migration [10]. The RAGE has emerged as a central regulator of vascular inflammation and atherosclerosis [11]. The soluble form of RAGE (sRAGE) is a product of both alternative splicing of RAGE mRNA and cleavage of membrane-bound RAGE [12]. The axis of the AGEs and its receptor’s RAGE is involved in arterial stiffness and consequently in hypertension [13]. Accumulating evidence has suggested that sRAGE and AGEs seem to be contributing factors in the development of atherosclerosis, coronary artery disease [14], and peripheral artery diseases, irrespective of either renal impairment or diabetes mellitus (DM) [15]. Even though the elevated levels of sRAGE in diabetic patients have been linked to coronary artery disease [16], and AGEs have been designated to be a predictor for mortality because of cardiovascular diseases [17,18], recently sRAGE has been considered as an oxidative stress biomarker as well [19]. In addition, sRAGE is used as a biomarker for reflecting the RAGE activity. sRAGE was shown to be elevated after the myocardial infarction and the increased sRAGE levels were associated with poor in-hospital prognosis [20]. However, the role of sRAGE within the CVD of Hemodialysis (HD) patients, remains uncertain. A previous cross-sectional study [21] reported that sRAGE was negatively correlated with CVD in HD, and that they may be acting as a decoy receptor in diabetics [22], or in non-diabetics [23]. On the other hand, sRAGE was reported to be unrelated to CVD in the chronic kidney disease (CKD) patients that were receiving dialysis [24]. Contrasting data demonstrating the association of CVD risk and diseases with sRAGE have been elucidated. Therefore, our aim was to assess the correlation of AGEs and their soluble receptors; sRAGE among HD patients, in relation to the underlying cause of CKD, whether diabetes mellitus; or any other cause, as well as to assess their association with CVD and its risk factors among HD patients. 

## 2. Subjects and Methods

This study involved 210 subjects, 130 of whom were hemodialysis patients that had been recruited from Hemodialysis Unit-ALNOOR hospital (Makkah, KSA), from January 2015 to December 2016. This study was approved by the ethical committee of the Institute of Scientific Research and AL-Noor hospital, with signed consent from all of the participants (Grant N: 43409001; ethical code (4327) at 1-12015).

All of the patients were on regular HD for three sessions/week, free from any condition that may have interfered with the serum levels of the AGEs and RAGEs—such as active liver disease, infection or underlying malignancy, protein-losing enteropathies, severe heart failure, or any acute illness within three months of enrolment. Furthermore, the patients who had missed more than one dialysis session per month were excluded from the study. There were 80 age and gender matched normotensive, apparently healthy individuals that were selected as a control group, with no evidence of CKD, as proved by an estimated glomerular filtration rate (eGFR) above 60 mL/min, according to the National Kidney Guidelines [25]. In addition, they had no evidence of ischemic heart disease, as proven by their history, with no clinical evidence of previous strokes [26]. According to the underlying causes of end stage renal disease (ESRD), the HD patients were categorized into diabetic HD patients—41 (31.5%)—and non-diabetic HD patients—89 (68.5%). The underlying causal-disorders for the nondiabetic ESRD were hypertensive nephropathy as well as other causes like focal segmental glomerulosclerosis, amyloidosis, and atrophic kidney. Type 2 diabetes mellitus is defined as a fasting blood glucose (FBG) ≥ 126 mg/dL, a two hour post prandial glucose (2HPPG) ≥ 200 mg/dL, glycosylated hemoglobin (HbA1C) ≥ 6.5%, or a random plasma glucose of ≥ 200 mg/dL in a patient with classic symptoms of hyperglycemia [27] or a patient that is on anti-diabetic therapy. Seventy four (36.1%) of the HD patients were categorized as having CVD, based on the criteria verified by [28], namely: if they had a past history of ischemic heart disease, symptomatic stroke verified by computed tomography and/or magnetic resonance imaging, or symptomatic peripheral vascular disease verified by lower-limb angiography and/or computed tomography angiography. Based on the above criteria, the HD patients were stratified into four sub-groups according to the associated CVD comorbidities, as follows: 16.2% (21) were diabetic patients with CVD (DM^+^CVD^+^), 15.4% (20) were diabetic patients without CVD (DM^+^CVD^−^), 20% (26) had CVD without DM (DM^−^CVD^+^) and the remainder of them—63 (48.5%)—had no associated comorbidity (DM^−^CVD^−^). Arterial hypertension was defined if the blood pressure was ≥140/90 mm Hg and/or whether the patients were on anti-hypertensive medication [26]. All of the cases and controls who had participated in our study were clearly informed about the aim of the study and they had agreed and signed a written informed consent, which had been approved by the UQU and Al-Noor specialized hospital ethical committees. Their anonymity was maintained as well. For all of the cases and healthy controls, a complete history and comprehensive questionnaires were used to collect socio-demographic data, medical information, and clinical data. Venous blood samples were collected and centrifuged for separation of the serum, which was stored at −80 °C, until analysis. Renal function tests (serum urea, creatinine, and uric acid), lipid profile (total cholesterol (TC), triglycerides (TG), low density lipoprotein cholesterol (LDLc) and high density lipoprotein cholesterol (HDLc)), and FBG were assessed by standard colorimetric techniques, according to the manufacturer’s instructions. Hypercholesterolemia was defined as a plasma level of the total cholesterol >200 mg/dL and the low-density lipoprotein cholesterol >129 mg/dL LDLc [29]. In addition, the serum albumin was assessed by immunochemical methods that were based on turbidimetry–nephelometry principles. The glycemic control was measured using HbA1c that utilized a high-performance liquid chromatography (HPLC). An assessment of the serum levels of the advanced glycation end products (AGEs), as described by [30], and the soluble receptors for the advanced glycation end products (sRAGE), as described by [31], was done using the enzyme linked immunosorbent assay (ELISA). Both were supplied by my-Biosource (San Diego, CA, USA).

## 3. Statistical Analysis

Statistical analysis was done using SPSS software, version 16 (Microsoft Corporation, Redmond, WA, USA). The categorical data have been presented as percentages and the continuous variables as means ± standard deviation for the parametric data, and as median with range for non-parametric data. The comparisons between the groups were calculated using the Chi-Square test with *p* for categorical variables, and by *t*-test and Mann–Whitney test for the continuous variables. The ANOVA post hoc test and Kruskal–Wallis test were used when comparing more than two continuous variables. The Spearman rank correlation analysis was used to determine the associations between the AGEs and sRAGE, with selected parameters. Linear regression analysis was used to assess the independent predictors of AGEs and sRAGE. *p* Values < 0.05 were considered significant. 

## 4. Results

The clinical and biochemical data of the study population (diabetic and non-diabetic HD group vs. control group) were summarized in Table 1 and Table 2. The HD sub-groups and control group were comparable for age (*p* = 0.16), gender (*p* = 0.17), and smoking habit (*p* = 0.29), as shown in Table 1. As expected, the HD patients, either diabetic or non-diabetic, had significantly higher urea (*p* = 0.0001) and creatinine (*p* = 0.0001), and lower albumin (*p* = 0.0001) and hemoglobin (Hb, *p* = 0.0001), compared with the control group (Table 1). The diabetic subgroup was older and was poorly controlled; the FBG was 196.1 ± 39.99 vs. 92.27 ± 6.94 and 91.23 ± 4.46, and the HbA1C was 8.21 ± 0.45 vs. 5.24 ± 0.22 and 5.25 ± 0.39, with a high prevalence of CVD, 51.2% vs. 29.2% (*p* = 0.015). Of the diabetic patients, 46.5% received metformin, while 41.5% and 12.2% received oral hypoglycemic drugs (OHA) and combined therapy (OHA and insulin), respectively. Although the diabetic patients were older, however, they matched for the age, duration of HD, and renal function tests (*p* = 0.94, 0.22, and 0.84, respectively) (Table 1). Moreover, the HD patients with DM had significantly increased LDLc (114 ± 18.21 vs. 105.54 ± 29.19) (*p* = 0.002), which indicated a high prevalence of hypercholesterolemia (46.3% vs. 25%), compared with the non-diabetic HD patients (*p* = 0.013). Moreover, the TG was significantly higher in the diabetic HD patients (166.76 ± 44.49), mainly those with CVD (184.14 ± 47.1), as compared with the other subgroups (*p* = 0.003), while the albumin level was significantly decreased in the diabetic HD patients, mainly those without CVD, compared with the other subgroups (*p* = 0.0001), Table 2. The diabetic HD and non-diabetic HD patients showed a significant increase of systolic blood pressure (SBP) compared with the control group, 142.32 ± 11.24 and 141.69 ± 8.08 vs. 119.75 ± 6.09, respectively, and the DBP, 91.09 ± 8.08 and 91.18 ± 5.44 vs. 80.25 ± 4.52, respectively (*p* = 0.0001) (Table 1). The highest value for both the SBP and diastolic blood pressure (DBP) was observed in the HD patients with CVD, for either diabetic or non-diabetic patients, 151.19 ± 6.1 and 148.46 ± 4.64, and 94.9 ± 6.61 and 94.62 ± 3.14, as compared with the other HD patients with no history of CVD (*p* = 0.0001), as shown in Table 2. In addition, the TC, 207.67 ± 31.08 and 190.19 ± 46.51 vs. 138.3 ± 25.61 and 159.76 ± 39.4, and LDLc, 126.05 ± 10.96 and 129.81 ± 34.69 vs. 100.95 ± 16.5 and 94.73 ± 16.92, were significantly increased in the HD patients with CVD, compared with those with no history of CVD (*p* = 0.0001), which indicated the significant prevalence of hypercholesterolemia among them, 81% and 61.5% vs. 10% and 11% (*p* = 0.0001) (Table 2). The serum levels of both the AGEs, 18.92 ± 4.15 vs. 16.89 ± 2.99, and sRAGE, 8.23 ± 2.34 vs. 3.01 ± 1.54, were significantly different in the diabetic, 19.3 ± 4.69 and 8.97 ± 1.53, and non-diabetic, 18.75 ± 3.89; and 7.89 ± 2.57, HD patients, and the non-CKD control group, 16.89 ± 2.99 and 3.01 ± 1.53, with a post hoc test significance between the diabetic and non-diabetic HD patients compared to the control group (*p* = 0.001 and *p* = 0.0001, respectively). Moreover, the sRAGE revealed a post hoc significance (*p* = 0.01), among HD regarding DM, but not for AGE (Table 3). Regarding the associated CVD, the median level of the AGEs levels was higher in the non-diabetic HD patients with CVD than those with no associated comorbidities (*p* = 0.011) (Table 3). Moreover, the sRAGE levels were significantly increased in the HD patients with CVD (Table 3), whether the underlying cause of renal impairment was DM or not (*p* = 0.015). This was proved by the higher prevalence of CVD, 39.1% *p* = 0.028 and 39.5% *p* = 0.033, among the non-diabetic HD patients with sRAGE > 9.23 (median) and AGEs > 17.9 (median), respectively (Table 4). Furthermore, the diabetic HD patients with AGEs > 17.9 (median) exhibited a higher prevalence of CVD (36% *p* = 0.016) and hypercholesterolemia (28%; *p* = 0.004) (Table 5). On the other hand, the diabetic HD patients with sRAGE > 9.23 (median) were anemic (50%; *p* = 0.008), as presented in Table 4. In addition, the circulating sRAGE was positively correlated with the poor control of DM (HbA1C; r = 0.24, *p* = 0.007) and hyperglycemia (r = 0.25, *p* = 0.004), mainly in those with sRAGE >9.23 (median), that reveled hypoalbuminemia (r = −0.32, *p* = 0.009), while the higher AGEs >17.9 median was associated with impaired renal function—UREA (r = 0.24, *p* = 0.044) and creatinine (r = 0.03 *p* = 0.012)—and low HDLc (not shown). Among the diabetic HD or non-diabetic patients, the univariable linear regression analysis (enter method) of the potential determinants of CVD revealed the AGEs (β (Beta Coefficient) = 0.17, *p* = 0.017 and β = 0.21, *p* = 0.023), sRAGEs (β = 0.15, *p* = 0.048 and β = 0.21, *p* = 0.035), and SBP (β = 0.62, *p* = 0.0001 and β = 0.559, *p* = 0.0001). However, the TC (β = 0.49, *p* = 0.0001) and urea (β = 0.18, *p* = 0.014) remained significantly associated with CVD, among the diabetic HD patients (Table 6). Thus, our study showed that the HD patients displayed significantly higher serum AGEs with a concomitant increase in the circulating sRAGE levels, mainly in diabetic HD patients, which were significantly associated with CVD (independent predictors) and the associated CV risk factors (hypercholesterolemia), regardless of underlying DM.

## 5. Discussion

The effects of the binding of the AGEs with their receptors’ RAGE, induced oxidative stress, inflammation, and extracellular matrix accumulation were translated into accelerated plaque formation and atherogenesis. Experimentally, the AGEs that were bound to the sRAGE prevented the proinflammatory effects by acting as a decoy receptor. However, the measurement of the sRAGE was a predictor of higher atherosclerotic or mortality risk. However, the issue remained to be tested in a specifically designed clinical trial. In this study, the HD patients displayed increased serum AGEs with a concomitant increase in their circulating soluble receptors (sRAGE), mainly among the diabetic HD patients that might have been as a result of both the increased production and reduced renal clearance. The increased AGEs were significantly associated with renal dysfunction (blood urea). Interestingly, the sRAGE were not only significantly related to diabetes that was associated with renal diseases, but it was also associated with cardiovascular diseases and their risk factors, regardless of DM. AGEs were significantly increased in the patients with a diminished renal function, even in the absence of hyperglycemia. They showed a non-significant difference among the uremic subjects with or without diabetes. In addition, the AGEs contributed to the progression of kidney disease in non-diabetic nephropathy, the possible mechanisms for this included binding to the RAGE, inducing oxidative stress, endothelial dysfunction, inflammation, and podocyte injury [32]. Another study reported significant increases of AGEs in HD patients with cardiovascular diseases, compared with the non-CKD group [33]. This implied that the increased accumulation of AGEs in the subjects with uremia might have been because of the increased oxidative stress, rather than the increased glucose burden alone. In contrast, Nazratun et al. [34] found that the patients with diabetic ESRD showed a higher immunohistochemical staining percentage of pentosidine (an AGE marker of blood vessels’ biopsies) than the non-diabetic ESRD or healthy subjects, which could have explained the controversial association of DM with the increased AGEs in the setting of uremia. In accordance to our results, numerous studies have demonstrated the strong association between the circulating levels of sRAGE and, not only the existence of renal impairment, but also the severity of kidney damage in patients with type 2 diabetes [35,36,37,38]. Conversely, another study denied such an association and revealed a non-significant difference of sRAGE among the diabetic and non-diabetic HD patients [21]. In addition, they reported a stronger positive staining in the ESRD patients, with hypertension as an additional comorbidity with diabetes.

Cardiovascular disease has become the leading cause of death in HD patients [39]. Our study confirmed this observation by eliciting a 36.1% prevalence of CVD among HD patients, mainly among diabetic HD patients, compared with non-diabetic HD patients (51.2% vs. 29.2%). Understanding the mechanisms that are involved in the development of CVD and the identification of biomarkers have been important steps in reducing the cardiovascular mortality in patients with CKD or in HD patients. The activation of RAGE by AGEs and other ligands has led to the attenuation of inflammatory response, oxidative stress, and apoptosis, which may have contributed to a variety of microvascular and macrovascular complications [40] in diabetes and chronic kidney diseases [9]. An inflammatory response via the AGEs’ interaction with RAGE would have triggered the release of inflammatory mediators and the activation of a nuclear factor-kappa B, with a concomitant induction of oxidative stress and atherosclerotic processes [40]. Daffu et al. (2013) suggested that the high serum AGEs were associated with vascular endothelial damage and accelerated cardiovascular morbidity, with an increased mortality in CKD patients [41]. Similarly, Furuya et al. (2015) showed that AGE accumulation was related to the chronic complications of DM and ESRD and was a significant independent risk factor for de novo CVD, according to the multivariate logistic analyses [42]. However, the existence of cardiovascular disease did not reveal a statistically significant contribution to the AGEs levels in our dialysis patients in comparison with the other study. This discrepancy could be related to a small sample size of the diabetic patients who were younger. Furthermore, Schwedler et al. (2002) did not find any association between the high AGEs’ levels and mortality [43]. Therefore, the significantly elevated AGEs in the uremia group were as a result of renal failure rather than DM or CVD. Lately, the serum levels of sRAGE were found to be significantly associated with the incidence of diabetic nephropathy, poor prognosis, and survival [44]. There were contrasting data demonstrating the association of CVD risk and diseases with sRAGE. The role of sRAGE as a vascular biomarker has remained uncertain, since a number of factors could have influenced its serum levels. A cross-sectional study previously described an association between the circulating levels of the sRAGE, atherosclerosis burden, and CVD risk in type 2 diabetes [45]. They reported an elevated serum level of sRAGE in type 2 diabetic patients with coronary artery disease or with an atherosclerotic burden. The risk for CVD events was higher as the serum sRAGE levels were increased in the studied diabetic patients. Similarly, our study revealed that sRAGE are significantly associated with CVD and in the HD patients, regardless of whether diabetes was an underlying cause of renal impairment. Conversely, in HD patients, Kim et al. (2013) showed a negative independent association of sRAGE with vascular calcification and cardiovascular disease, even after the adjustment of the important confounding factors [21]. In the Atherosclerosis Risk in Communities (ARIC) study, 1201 apparently healthy participants were followed for 18 years. The authors observed that the plasma levels of sRAGE were inversely associated with the risk for coronary heart disease (hazard ratio 1.82 [95% CI 1.17–2.84]), diabetes (1.64 [1.10–2.44]), and mortality (1.72 [1.11–2.64]). They concluded that low levels of sRAGE were a marker of risk for future chronic disease and mortality and might have represented a predictor of an inflammatory state [46]. The sRAGE was binded to the RAGE legends and inhibits in a competitive manner and the ligand/RAGE interaction in order to block the adverse effects of the RAGE signaling. The sRAGE prevented adverse effects of RAGE, such as diabetic atherosclerosis, and protected against atherosclerotic cardiovascular events [47]. Lately, some studies on chronic kidney disease (CKD) patients that were undergoing dialysis revealed that sRAGE was not related to CVD [24,48]. However, Villegas-Rodríguez et al. (2016) found a significant association between the AGE–RAGE axis markers, especially the sRAGE, with several non-invasive markers of cardiovascular-disease risk in the population of patients with newly diagnosed diabetes type 2 [49]. It was also suggested that sRAGE could be a biomarker for a RAGE-mediated disease, especially vascular-disease [12]. Furthermore, the sRAGE concentration and the genetic influence on sRAGEs levels might have partly explained the contradictions that have been seen between different the studies [50]. 

## 6. Conclusions 

Our results displayed a significantly higher serum of AGEs in HD patients with a concomitant increase in the circulating soluble form of the receptor for the advanced glycation end-products’ levels, mainly in HD patients with underlying diabetes. These were significantly associated with cardiovascular diseases (independent predictors) and cardiovascular risk factors (hypercholesterolemia), mainly sRAGEs, among HD patients, regardless of the underlying cause of chronic kidney diseases. This highlights the prognostic role of AGEs and sRAGE in HD patients, regardless of the underlying cause, in order to predict the risk of CVD. 

### Limitations

The present study has several limitations: Firstly, is the small sample size that was used to show the underlying cause-specific differences. Secondly, several other not controlled influences and circumstances might have led to an increase in AGEs and sRAGE levels. Further studies are recommended to evaluate whether persistent chronic diseases as CKD and DM could affect the role of AGEs and sRAGE as promising CVD biomarkers.

## Figures and Tables

**Table 1 medsci-06-00038-t001:** Demographic, clinical, and biochemical data between the studied groups.

Variables	Groups			*p* value
	Hemodialysis DM^+^	Hemodialysis DM^−^	Control	
	(41) 31.5%	(89) 68.5%	(80)	
Age (Years)	54.09 ± 10.1	50.54 ± 9.85	51.48 ± 9.74	0.16
Gender: Male No (%) Female No (%)	19 (46.3%) 22 (53.7%)	34 (38.2%) 55(61.8%)	42 (52.5%) 38 (47.5%)	0.17
HD Duration	2.85 ± 1.13	3.12 ± 1.2	-	0.22
Smokers No (%)	10 (24.4%)	14 (15.7%)	10 (12.5%)	0.24
Co-morbidity: CVD %	21 (51.2%)	26 (29.2%)	-	0.015 *
SBP (mmHg)	142.32 ± 11.24 145 (35)	141.69 ± 8.08 145 (35)	119.75 ± 6.09 120 (110–135)	*p*1 = 0.66 *p* 2 = 0.0001 *p* 3 = 0.0001
DBP (mmHg)	91.09 ± 8.08 90 (30)	91.18 ± 5.44 90 (20)	80.25 ± 4.52 80 (65–85)	*p* 1 = 0.42 *p* 2 = 0.0001 *p* 3 = 0.0001
Hypertension (HTN) No (%)	26 (63.4%)	69 (77.5%)	-	0.092 *
Hypercholesterolemia No (%)	19 (46.3%)	23 (25.8%)	-	0.02 *
FBG (mg/dL)	196.1 ± 39.99 184 (150)	92.27 ± 6.94 93 (28)	91.23 ± 4.46 90 (16)	*p* 1 = 0.0001 *p* 2 = 0.0001 *p* 3 = 0.41
HbA1C (%)	8.21 ± 0.45 8 (1.6)	5.24 ± 0.22 5.2 (0.8)	5.25 ± 0.39 5.3 (1.4)	*p* 1 = 0.0001 *p* 2 = 0.0001 *p* 3 = 0.95
DM TTT				
Metformin N (%) OHA N (%) Combined TTT N (%)	19 (46.3%) 17 (41.5%) 5 (12.2%)	- - -	- - -	
TC (mg/dL)	173.83 ± 45.02 158 (129)	168.65 ± 43.62 157 (193)	121.8 ± 20.24 113 (100)	*p* 1 = 0.55 *p* 2 = 0.0001 *p* 3 = 0.0001
TG (mg/dL)	166.76 ± 44.49 165 (163)	143 ± 54.43 136 (172)	99.1 ± 11.19 103 (34)	0.0001 •
LDLc (mg/dL)	113.8 ± 18.73 121(58)	104.98 ± 28.31 96 (122)	87.95 ± 15.88 91(44)	*p* 1 = 0.004 *p* 2 = 0.0001 *p* 3 = 0.0001
HDLc (mg/dL)	37.78 ± 7.23 38 (31)	38.1 ± 4.79 39 (25)	46.47 ± 3.32 46 (41–52)	*p* 1 = 0.49 *p* 2 = 0.0001 *p* 3 = 0.0001
Hemoglobin (Hb)	11.21 ± 1.29 11 (4.3)	10.98 ± 1.7 11 (6.9)	14.27 ± 1.26 14 (12–16)	0.0001 ◊
BUN (mg/dL)	109.95 ± 15.99 107 (52)	109.92 ± 16.99 102 (65)	26.4 ± 3.65 27 (13)	*p* 1 = 0.94 *p* 2 = 0.0001 *p* 3 = 0.0001
Creatinine (mg/dL)	9.77 ± 1.29 9.7 (6.3)	9.88 ± 1.74 9.5 (9.2)	0.89 ± 0.12 0.9 (0.4)	*p* 1 = 0.84 *p* 2 = 0.0001 *p* 3 = 0.0001
Uric acid (mg/dL)	6.26 ± 2.17 5.9 (10.2)	6.55 ± 1.78 6.2(8)	5.89 ± 0.96 6.1 (3.4)	0.19
Albumin (mg/dL)	3.15 ± 0.47	3.46 ± 0.37	4.46 ± 0.32	0.0001 *

Mann–Whietney test: *p*1 = HD DM^+^ vs. HD DM^−^; *p*2 = HD DM^+^ vs. control; and *p* 3 = HD DM^−^ vs. control. *—post hoc test: HD DM^+^ vs. HD DM^−^ vs. control (*p* = 0.0001); •—post hoc test: HD DM^+^ vs. HD DM^−^ vs. control (*p* = 0.007, 0.0001); ◊—post hoc test: HD DM^+^ & HD DM^−^ vs. control (*p* = 0.0001); HD—hemodialysis; DM—diabetes mellitus; SBP—systolic blood pressure; DBP—diastolic blood pressure; TC—total cholesterol; TG—triglycerides; HDLc—high density lipoprotein cholesterol; LDLc—low density lipoprotein cholesterol; BUN—blood urea nitrogen.

**Table 2 medsci-06-00038-t002:** Comparison of demographic and clinical data among hemodialysis sub-groups.

Variables	Hemodialysis Sub-Groups				*p* Value
	DM^+^CVD^+^ (21)	DM^+^CVD^−^ (20)	DM^−^CVD^+^ (26)	DM^−^CVD^−^ (63)	
Age	54.57 ± 10.15	53.6 ± 10.3	50.73 ± 8.49	50.46 ± 10.41	0.31
Gender: Male Female	6 (28.6%) 15 (71.4%)	13 (65%) 7 (35%)	8 (30.8%) 18 (69.2%)	26 (41.3%) 37 (58.7%)	0.07
Smokers N (%)	8 (38.1%)	2 (10%)	3 (11.5%)	11 (17.5%)	0.07
SBP	151.19 ± 6.1 150 (15)	133 ± 6.96 130 (20)	148.46 ± 4.64 150 (20)	138.89 ± 7.54 140 (25)	0.0001 •
DBP	94.9 ± 6.61 95 (20)	85 ± 3.63 85 (10)	94.62 ± 3.14 95 (10)	89.76 ± 5.57 90 (20)	0.0001 •
Hypercholesterolemia	17 (81%)	2 (10%)	16 (61.5%)	7 (11.1%)	0.0001
FBG	195.57 ± 30.93 200 (105)	196.6 ± 48.58 180.5 (150)	92.58 ± 5.85 93 (18)	92.14 ± 7.39 93 (28)	0.0001 *
HbA1C	8.33 ± 0.51 8.5 (1.6)	8.09 ± 0.34 8 (1.1)	5.28 ± 0.21 5.35 (0.6)	5.23 ± 0.22 5.2 (0.8)	0.0001 •
TC	207.67 ± 31.08 209 (93)	138.3 ± 25.61 132.5 (94)	190.19 ± 46.51 203 (135)	159.76 ± 39.4 155 (193)	0.0001 •
TG	184.14 ± 47.1	148.5 ± 33.89	158.5 ± 51.57	136.6 ± 54.69	0.003 ″
LDLc	126.05 ± 10.96 131 (35)	100.95 ± 16.5 97 (44)	129.81 ± 34.69 133.5 (121)	94.73 ± 16.92 89 (80)	0.0001 •
HDLc	38.71 ± 9.12 39 (31)	36.8 ± 4.51 38 (16)	36.46 ± 5.79 39 (17)	38.78 ± 4.18 39 (22)	0.23 •
Hb	11.07 ± 1.33	11.36 ± 1.27	10.43 ± 2.01	11.2 ± 1.5	0.15 •
Uric acid	6.96 ± 2.75 5.7 (9.2)	5.53 ± 0.92 6 (2.6)	6.11 ± 1.41 6.1 (4.8)	6.73 ± 1.89 6.2 (8)	0.15 •
Albumin	3.26 ± 0.41	3.03 ± 0.5	3.55 ± 0.44	3.42 ± 0.33	0.0001 • ◊

* Kruskal–Wallis test. • One-Way Anova Test/post hoc significance, ″ DM^+^CVD^+^ vs. DM^−^CVD^−^ (*p* = 0.003), ◊ DM^+^CVD^−^ vs. DM^−^CVD^+^ and DM^−^CVD^−^ (*p* = 0.005, 0.019); FBG—fasting blood glucose; HbA1C—glycosylated hemoglobin; Hb—hemoglobin.

**Table 3 medsci-06-00038-t003:** Comparison of biochemical data among the studied groups.

Group	AGEs	*p* Value	sRAGE	*p* Value
Control group	16.89 ± 2.98	0.001	3.01 ± 1.53	0.0001
	16.82 (11.1)		3 (3.8)	
All Hemodialysis groups	18.92 ± 4.15 17.9 (20.2)		8.23 ± 2.34 9.3 (9.6)	
Diabetic HD	19.3 ± 4.69	0.001 *	8.97 ± 1.53	0.0001 *^,^#
	18.5 (18)		9.6 (6.52)	
Non-diabetic HD	18.75 ± 3.89 17.8 (17.1)		7.89 ± 2.57 9.23 (9.6)	
Control	16.89 ± 2.99 16.82 (11.1)		3.01 ± 1.53 3 (3.8)	
CVD	19.49 ± 4.34	0.24	9.03 ± 1.54	0.003
	17.9 (18)		9.68 (6.43)	
Non-CVD	18.6 ± 4.03 17.9 (17.9)		7.77 ± 2.59 8.3 (9.6)	
Hemodialysis sub-groups				
DM^+^CVD^+^ (1)	18.55 ± 5.23	0.011 ″	8.97 ± 1.85	0.015 ″
	17.24 (18)		9.77 (6.4)	
DM^+^CVD^−^ (2)	20.09 ± 4.03 21.03 (13.9)		8.97 ± 1.15 9.1 (2.77)	
DM^−^CVD^+^ (3)	20.25 ± 3.38 19.6 (9.6)		9.09 ± 1.27 9.68 (3.83)	
DM^−^CVD^−^ (4)	18.13 ± 3.95 16.98 (17.1)		7.39 ± 2.81 8.3 (9.6)	

*—post hoc test significance between diabetic and non-diabetic HD patients, and the control group; #—post hoc test significance between the diabetic and non-diabetic HD patients; ″—Kruskal–Wallis test. AGEs—advanced glycation end-products; sRAGE—soluble receptor of advanced glycation end-products; post hoc significance revealed significant difference between CVD^+^ (3) vs. DM^−^CVD^−^(4), regarding the sRAGEs (*p* = 0.001) but not the AGEs (*p* = 0.078).

**Table 4 medsci-06-00038-t004:** Baseline characteristic of HD DM^−^ patients according to AGEs and sRAGE (median level).

Variable	AGEs < 17.9	AGEs > 17.9	*p* Value	sRAGE < 9.23	sRAGE > 9.23	*p* Value
Age						
Age <50 Age >50	22 (47.8%) 24 (52.2%)	29 (67.4%) 14 (32.6%)	0.049	19 (44.2%) 24 (55.8%)	32 (69.6%) 14 (30.4%)	0.013
Gender						
Male Female	21 (45.7%) 25 (54.3%)	13 (30.2%) 30 (69.8%)	0.1	17 (39.5%) 26 (60.5%)	17 (37%) 29 (63%)	0.48
Smokers	9 (19.6%)	75(11.6%)	0.23	4 (9.3%)	10 (21.7%)	0.09
Hypertension	35 (76.1%)	34 (79.1%)	0.5	32 (74.4%)	37 (80.4%)	0.3
Hypercholesterolemia	12 (26.1%)	11 (25.6%)	0.57	4 (9.3%)	19 (41.3%)	0.001
Comorbidity						
CVD/-CVD ^+^ACE/-ACE	9 (19.6%) 2 (4.3%)	17 (39.5%) 6(14%)	0.033 0.11	8 (18.6%) 4 (9.3%)	18 (39.1%) 4 (7.8%)	0.028 0.6
Hb % < 11	17 (37%)	21 (48.8%)	0.18	16 (37.2%)	22 (47.8%)	0.21

**Table 5 medsci-06-00038-t005:** Baseline characteristic of HD DM^+^ patients according to AGEs and sRAGE (median level).

Variable	AGEs < 17.9	AGEs > 17.9	*p* Value	sRAGE < 9.23	sRAGE > 9.23	*p* Value
Age						
Age < 50 Age > 50	4 (25%) 12 (75%)	10(40%) 15 (60%)	0.26	7 (36.8%) 12 (63.2%)	7 (31.8%) 15 (68.2%)	0.49
Gender						
Male Female	4 (25%) 12(75%)	15 (60%) 10 (40%)	0.03	6 (31.6%) 13 (68.4%)	13 (59.1%) 9 (40.9%)	0.07
Smokers	3 (18.8%)	7 (28%)	0.38	5 (26.3%)	5 (22.7%)	0.5
Hypertension	12 (75%)	14 (56%)	0.19	13 (68.4%)	13 (59.1%)	0.39
Hypercholesterolemia	12 (75%)	7 (28%)	0.004	8 (42.1%)	11 (50%)	0.4
Comorbidity						
CVD ^+^ACE	12 (75%) 2 (12.5%)	9 (36%) 5 (20%)	0.016 0.4	8 (42.1%) 2 (10.5%)	13 (59.1%) 5 (22.7%)	0.22 0.27
Hb % < 11	4 (25%)	9 (36%)	0.35	2 (10.5%)	11 (50%)	0.008

DM—diabetes mellitus; CVD—cardiovascular diseases; ACE— acute coronary event.

**Table 6 medsci-06-00038-t006:** Linear regression analysis to investigate independent factors associated with CVD in HD.

Variable	CVD		
	β	*p*	CI
HD DM^+^			
AGEs	0.17	0.017	0.04–0.034
sRAGEs	0.146	0.048	0.000–0.096
SBP	0.62	0.0001	0.02–0.036
TC	0.49	0.000	0.003–0.008
FBG	0.011	0.29	−0.001–0.004
HbA1C	0.03	0.78	−0.216–0.283
Urea	0.18	0.014	0.001–0.01
Albumin	−0.08	0.3	−0.27–0.09
HD DM^−^			
AGEs	0.21	0.023	0.003–0.046
sRAGEs	0.21	0.035	0.003–0.072
SBP	0.559	0.0001	0.021–0.042
TC	0.067	0.5	−0.002–0.003
FBG	−0.18	0.058	−0.024–0.000
HbA1C	0.11	0.23	−0.16–0.54
Urea	−0.12	0.19	−0.008–0.002
Albumin	0.011	0.9	−0.21–0.24

Predictors (constant): albumin, AGEs, sRAGEs, FBG, Urea, SBP, TC, and HbA1C; β (Beta coefficient. dependent variable: CVD.

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
