# Peer review of "Critical Appraisal of Advanced Glycation End Products (AGEs) and Circulating Soluble Receptors for Advanced Glycation End Products (sRAGE) as a Predictive Biomarkers for Cardiovascular Disease in Hemodialysis Patients"

_medsci, 2018, doi:10.3390/medsci6020038_

Reviewer 1 Report

The manuscript is interesting and overall well written. I just have few comments:

-A more integrated appraisal of the relevant literature would be appropriate to provide the context for the study:

In skeletal muscle advanced glycation end products (AGEs) inhibit insulin action and induce the formation of multimolecular complexes including the receptor for AGEs. J Biol Chem. 2008.

Long-term physical activity leads to a significant increase in serum sRAGE levels: a sign of decreased AGE-mediated inflammation due to physical activity? Heart Vessels. 2018

Advanced Glycation End Products, Oxidation Products, and Incident Cardiovascular Events in Patients With Type 2 Diabetes. Diabetes Care. 2018

-A short paragraph in the beginning of the Discussion identifying the main points that will be covered, following the major effects seen in the results section might help tie together the discussion better. Moreover, the discussion is unnecessarily long with a hodge-podge of several studies performed in this area that take away from the story at hand. This detracts from the authors' ability to convey the study’s main findings.

-The study strengths and limitations should be deeply addressed, taking into account sources of potential bias or imprecision: Discuss both direction and magnitude of any potential bias.

Author Response

Reviewer letter

Reviewer 1

1.     English language and style

( ) Extensive editing of English language and style required 
( ) Moderate English changes required 
(x) English language and style are fine/minor spell check required 
( ) I don't feel qualified to judge about the English language and style 

This manuscript was automatically checked for spelling correction

 Yes

Can be improved

Must be improved

Not applicable

Does the introduction provide sufficient background and include all   relevant references?

( )

( )

(x)

( )

A more integrated appraisal of the relevant literature appropriate to provide the context for the study were implemented in our study

In skeletal muscle advanced glycation end products (AGEs) inhibit insulin action and induce the formation of multimolecular complexes including the receptor for AGEs. J Biol Chem. 2008.

Long-term physical activity leads to a significant increase in serum sRAGE levels: a sign of decreased AGE-mediated inflammation due to physical activity? Heart Vessels. 2018

Advanced Glycation End Products, Oxidation Products, and Incident Cardiovascular Events in Patients With Type 2 Diabetes. Diabetes Care. 2018

Is the research design appropriate?

(x)

( )

( )

( )

Are the methods adequately described?

(x)

( )

( )

( )

Are the results clearly presented?

(x)

( )

( )

( )

Are the conclusions   supported by the results?

( )

(x)

( )

( )

We have modify the conclusion to be supported by the results

A short paragraph in the beginning of the Discussion identifying the main points that will be covered, following the major effects seen in the results section might help tie together the discussion better

A short paragraph was implemented in the beginning of the Discussion identifying the main points that will be covered, following the major effects seen in the results section might help tie together the discussion better

The study strengths and limitations should be deeply addressed, taking into account sources of potential bias or imprecision: Discuss both direction and magnitude of any potential bias

The study strengths and limitations have been deeply addressed

Reviewer 2 Report

This is a cross-sectional study trying to evaluate the association of serum AGE, sRAGE with CVD and risk factors of CVD in a group of HD patients. The manuscript is well-written with an appropriate healthy control. However, the way that the authors analyzed the data could be misleading. To conclude that sRAGE were significantly associated with CVD, a few issues should be clarified.

1.      In table 1, duration of hemodialysis and any therapies for DM should be provided.

2.      In table 3, post-hoc comparisons of serum levels of AGEs and sRAGE in hemodialysis sub-groups were not clearly demonstrated. Any significant difference between DM-CVD+ (3) vs. DM-CVD-(4)?

3.      Table 4, 5 covered similar results and seemed redundant. I suggested combine the analysis (AGEs < 17.9 vs. > 17.9; sRAGE<9.23 vs.="" srage="">9.23) in HD patients with/without DM, respectively. Since CVD and ACE were analyzed separately, how about stroke and peripheral arterial disease?

4.      In Table 6, the regression analysis put AGE and sRAGE as the dependent variables. However, to study whether AGE or sRAGE could be useful biomarkers for CVD, the authors should analyze CVD as dependent variable and include AGE, sRAGE, SBP, TC, FBG… as the independent variable. In addition, how the regression analysis was done (univariable vs. multivariable, enter, stepwise or forward?) should be provided in the manuscript. Since the levels of AGE, sRAGE are highly associated with DM, the analysis should be performed in patients with and without DM, separately.

Author Response

Reviewer 2

Open Review

English language and style

( ) Extensive editing of English language and style required 
( ) Moderate English changes required 
(x) English language and style are fine/minor spell check required 
( ) I don't feel qualified to judge about the English language and style 

This manuscript was automatically checked for spelling correction

Yes

Can be improved

Must be improved

Not applicable

Does the introduction provide sufficient background and include all   relevant references?

(x)

( )

( )

( )

Is the research design appropriate?

( )

(x)

( )

( )

Are the methods adequately described?

( )

( )

(x)

( )

Are the results clearly presented?

( )

( )

(x)

( )

Are the conclusions supported by the results?

( )

(x)

( )

( )

Comments and Suggestions for Authors

This is a cross-sectional study trying to evaluate the association of serum AGE, sRAGE with CVD and risk factors of CVD in a group of HD patients. The manuscript is well-written with an appropriate healthy control. However, the way that the authors analyzed the data could be misleading. To conclude that sRAGE were significantly associated with CVD, a few issues should be clarified.

1.     In table 1, duration of hemodialysis and any therapies for DM should be provided.

The duration of hemodialysis and any therapies for DM were provided in Table 1

2.     In table 3, post-hoc comparisons of serum levels of AGEs and sRAGE in hemodialysis sub-groups were not clearly demonstrated. Any significant difference between DM-CVD+ (3) vs. DM-CVD-(4)?

   In table 3, post-hoc comparisons of serum levels of AGEs and sRAGE in hemodialysis sub-groups were clearly demonstrated.

3.     Table 4, 5 covered similar results and seemed redundant. I suggested combine the analysis (AGEs < 17.9 vs. > 17.9; sRAGE<9.23 vs.="" srage="">9.23) in HD patients with/without DM, respectively. Since CVD and ACE were analyzed separately, how about stroke and peripheral arterial disease?

The results of  (AGEs < 17.9 vs. > 17.9; sRAGE<9.23 vs.="" srage="">9.23) in HD patients with/without DM were analysed in Table 4 and Table 5

4.      In Table 6, the regression analysis put AGE and sRAGE as the dependent variables. However, to study whether AGE or sRAGE could be useful biomarkers for CVD, the authors should analyze CVD as dependent variable and include AGE, sRAGE, SBP, TC, FBG… as the independent variable. In addition, how the regression analysis was done (univariable vs. multivariable, enter, stepwise or forward?) should be provided in the manuscript. Since the levels of AGE, sRAGE are highly associated with DM, the analysis should be performed in patients with and without DM, separately.

In Table 6, the regression analysis, CVD was analysed as dependent variable and include AGE, sRAGE, SBP, TC, FBG… as the independent variable.

In addition, how the regression analysis was done (univariable  and enter) provided in the manuscript.

 Since the levels of AGE, sRAGE are highly associated with DM, the analysis was performed in patients with and without DM, separately.